# Alkaline Soil Degradation and Crop Safety of 5-Substituted Chlorsulfuron Derivatives

**DOI:** 10.3390/molecules27103318

**Published:** 2022-05-21

**Authors:** Lei Wu, Xue-Wen Hua, Yong-Hong Li, Zhong-Wen Wang, Sha Zhou, Zheng-Ming Li

**Affiliations:** 1State Key Laboratory of Elemento-Organic Chemistry, College of Chemistry, Nankai University, Tianjin 300071, China; wl836929871@163.com (L.W.); bioassay@nankai.edu.cn (Y.-H.L.); wzwrj@nankai.edu.cn (Z.-W.W.); 2College of Agriculture, Liaocheng University, Liaocheng 252059, China; huaxuewen900@163.com

**Keywords:** sulfonylurea herbicides, chlorsulfuron, alkaline soil degradation, DT_50_

## Abstract

Sulfonylurea herbicides can lead to serious weed resistance due to their long degradation times and large-scale applications. This is especially true for chlorsulfuron, a widely used acetolactate synthase inhibitor used around the world. Its persistence in soil often affects the growth of crop seedlings in the following crop rotation, and leads to serious environmental pollution all over the world. Our research goal is to obtain chlorsulfuron-derived herbicides with high herbicidal activities, fast degradation times, as well as good crop safety. On account of the slow natural degradation of chlorsulfuron in alkaline soil, based on the previously reported results in acidic soil, the degradation behaviours of 5-substituted chlorsulfuron analogues (**L101**–**L107**) were investigated in a soil with pH 8.39. The experimental data indicated that 5-substituted chlorsulfuron compounds could accelerate degradation rates in alkaline soil, and thus, highlighted the potential for rational controllable degradation in soil. The degradation rates of these chlorsulfuron derivatives were accelerated by 1.84–77.22-fold, compared to chlorsulfuron, and exhibited excellent crop safety in wheat and corn (through pre-emergence treatment). In combination with bioassay activities, acidic and alkaline soil degradation, and crop safety, it was concluded that compounds **L104** and **L107**, with ethyl or methyl groups, are potential green sulfonylurea herbicides for pre-emergence treatment on wheat and corn. This paper provides a reference for the further design of new sulfonylurea herbicides with high herbicidal activity, fast, controllable degradation rates, and high crop safety.

## 1. Introduction

Herbicides belonging to the group of organic compounds called sulfonylureas have extensively been used since 1987, as they offer the advantages of strong activity, low toxicity, low dosage, and are safe for mammals [1]. Sulfonylurea herbicides target acetolactate synthase (ALS) in weeds, which catalyses the synthesis of branched-chain amino acids in plants and microbes [2]. Their mode of action is to inhibit the biosynthesis of leucine, isoleucine, and valine, which results in growth inhibition and leads to the death of weeds [3].

Sulfonylurea herbicides can lead to serious weed resistance due to their long degradation times and large-scale application on croplands [4]. Blair et al., reported that the residues of chlorsulfuron had caused great harm to the sugarbeet [5]. Anderson et al., reported that three years after the application of chlorsulfuron in a wheat field, its soil residue still endangered the growth of sensitive crops such as corn and sunflower [6]. Walker et al., reported that the persistence of chlorsulfuron in soil could affect the growth of sugarbeet and lettuce sown approximately one year after application [7]. A study by Rother et al., demonstrated that the application of chlorsulfuron caused serious crop damage to flax and lentils in neutral soil [8].

Continuous planting of crops within a year on the same plot is a common practice in China [9]. Chlorsulfuron is a classical sulfonylurea herbicide that can seriously endanger the normal growth of crop seedlings in the following cropping season (with herbicide application in the prior cropping/harvesting cycle) due to its persistence in soil; it has been banned in China since 2014 [10].

Studies have shown that the pH of the soil can influence the degradation rates of sulfonylurea herbicides in soils [6]. Anderson et al., reported that the DT_50_ (half-life of degradation) of chlorsulfuron was 37.0 days^.^ In soil with pH 5.7 [6]. A study by Fredrickson et al., demonstrated that the DT_50_ of chlorsulfuron was 1.9 weeks in soil with pH 5.6, and 2.7 weeks in soil with pH 6.3 [11]. In a previous study, Hua et al., observed that the introduction of electron-donating and -withdrawing substituents onto the fifth position of the benzene ring in chlorsulfuron could influence degradation rates at pH 5.41 [12,13]. Meng et al., introduced dialkylamino substituents on the same position in chlorsulfuron to illustrate the relationship between structures and DT_50_ in soil with pH 5.52 [14].

The degradation speed under alkaline conditions was slower than that under acidic conditions. Chlorsulfuron is very persistent in alkaline soil with a DT_50_ of 144 days at pH 8.1 at 20 °C [15]. Walker et al., found that the DT_50_ of chlorsulfuron was 56.0 days in sandy soil (at 6% soil moisture) with pH 7.1 [7]. In our previous study, Zhou et al. found that 5-dialkylamino-substituted groups on the benzene ring of chlorsulfuron could greatly accelerate degradation under alkaline conditions [16,17]. However, Zhou found that 5-dialkylamino-substituted chlorsulfuron derivatives were less safe for use on wheat and corn through post-emergence treatment [17]. Moreover, the degradation rates of 5-dialkylamino-substituted chlorsulfuron derivatives were too fast for practical field use [16,17].

Therefore, it is urgent to devise innovative herbicides with high crop safety that permit fast and controllable degradation in alkaline soils, which are suitable for farming modes with different fallow periods.

Previous studies have shown that 5-dialkylamino-substituted chlorsulfuron derivatives could greatly accelerate their degradation rates in acidic and alkaline soils [14,17,18]. It was found that most 5-substituted chlorsulfuron derivatives maintained high herbicidal activities (Figure 1) [12,13]. However, their alkaline soil degradation and crop safety on wheat and corn have not been studied systematically.

It was speculated that 5-substituents on the benzene ring might accelerate degradation in alkaline conditions and improve safety on sensitive crops. In order to obtain sulfonylurea herbicides with fast, controllable degradation rates and high crop safety, 5-substituted chlorsulfuron derivatives (**L101**–**L107**) that maintain high herbicidal activities were selected and systematically investigated under alkaline conditions. Alkaline soil with pH 8.39 was selected from Cangzhou (in the Hebei Province of China) [18]. These compounds were studied with chlorsulfuron as the control. By combining information on their structures, bioassay activities, acidic and alkaline soil degradation rates, and crop safety, an insight into the multi-factor relationship was established for the innovation of novel green sulfonylurea herbicides.

## 2. Materials and Methods

### 2.1. Instruments and Materials

All reagents used were of chromatographic grade for HPLC (high-performance liquid chromatography) and reagents for reaction were of analytical grade. SHIMADZU LC-20AT (SHIMADZU Co., Tokyo, Japan) was used for absorption studies, and the data were analysed on a desktop computer (Vostro 3670, Dell, Round Rock, TX, USA). The wavelength was detected by UV (ultraviolet-visible spectrophotometer; TU-1810, Persee General Analysis Co., Beijing, China). Thermo Scientific Legend Mach 1.6 R centrifuge (Thermo Fisher Scientific Inc., Waltham, MA, USA) was used to separate the soil and organic layers. The biochemical incubator (Boxun Industrial Co., Shanghai, China) was used to perform the degradation experiment.

### 2.2. Compounds ***L101***–***L107***

The structure of 5-substituted chlorsulfuron compounds is shown in Figure 2.

The procedures for synthesising 5-substituted sulfonylurea analogues are shown in Figure 3; these were reported in our previous papers, as well [12,13,14].

### 2.3. Soil Degradation Assay

Soil degradation steps have been reported in detail by some previous studies [12,13,14,16,17,18,19,20], and are described here, briefly. Firstly, the selection of a suitable soil sample was carried out. Alkaline topsoil was collected from the upper layer (0–25 cm) from fresh farmland, and then sifted through a 2 mm sieve after air-drying under shade. The properties of this alkaline soil are listed in Table 1.

Secondly, analysis of the target compounds by HPLC was performed according to the Chinese National Standard GB/T 16631-2008 [19]. Chromatographical grade methanol, acetonitrile, and H_3_PO_4_ (aq) (pH 3.0) were used as the mobile phase. The wavelength of the target compounds was 235 nm. Standard curves were established for them at 20 °C, with the injection volume of 10 μL, and the concentration range was between 200 μg·mL^−1^ and 0.025 μg·mL^−1^. The retention time was no less than 10 min.

Thirdly, the recovery rate experiment was performed. The concentrations of the test compounds were 0.5, 2, and 5 mg·kg^−1^ (in acetonitrile) for 20 g of soil. Three concentrations for all samples were tested 5 times, and the recovery rates were expected to range from 70 to 110% with their coefficient of variation being <5% [20]. Analytical data on the verification of recovery rates at various concentrations are listed in Table 2.

Finally, the degradation culture of the samples was prepared. The concentration of each target sample was 5 mg·kg^−1^, in soil. The water-holding capacity of the soil was regulated by adding 3.5 mL of water. The samples were cultivated at 25 ± 1 °C with 80% humidity in the dark in a biochemical incubator. Soil samples were collected in triplicates at six different times. DT_50_ values were obtained by DT_50_ = ln(2)/k. Details of these steps can be found in the Appendix A description.

### 2.4. Crop Safety Assay

Chlorsulfuron is a popular sulfonylurea herbicide applied to wheat fields, worldwide. However, its persistence in soil often affects the growth of crop seedlings in the following crop rotation after it has been applied to one crop. Under the Chinese special crop rotation model, corn is one of the crops planted after wheat. In this study, corn was selected as the next crop to be planted, after wheat. The safety of target compounds on wheat (Xinong 529) and corn (Xindan 66) was studied with chlorsulfuron as the control. The method of culturing the plants has been reported previously [14,16,17,18].

Methods of plant cultivation: We filled the artificially mixed soil (vermiculite, loam, fertilizer soil (*V/V/V* = 1:1:1)) into a 7.0 cm-diameter paper cup (250 mL). Crop seeds (0.6 cm deep) were planted in mixed soil. Before the plants sprouted, plastic wrap was used to cover the cups to keep them moist. The plants were cultivated in a greenhouse at 25 ± 1 °C. Plants were watered regularly to ensure normal growth.

Wheat safety assay: Pot trials were used to test the target compounds under pre- and post-emergence at 30 and 60 g·ha^−1^. For pre-emergence treatment, the fresh weight of the cover crops was determined after 22 days. For post-emergence treatment, the fresh weight of the cover crops was determined when wheat grew to the four-leaf stage. The fresh weight of the cover crops was measured 28 days after spraying.

In the case of corn, the detailed crop safety assay was consistent with that of wheat. For pre-emergence treatment, the fresh weight of the cover crops was determined after 16 days. For post-emergence treatment, the safety assay started when wheat grew to the three-leaf stage. The fresh weight of the cover crops was determined 23 days after spraying.

The fresh weight of the cover crops was measured after several days, and the inhibition rates of the fresh weight were used to represent the safety of the crops. The data were analyzed through Duncan multiple comparison by SPSS 22.0.

## 3. Results and Discussion

### 3.1. Soil Degradation

With chlorsulfuron as the positive control, degradation of this series of target compounds was systematically investigated in soil with pH 8.39. The kinetic parameters are listed in Table 3.

As shown in Table 3, the DT_50_ of chlorsulfuron was 157.53 days and the DT_50_ values of **L101**–**L107** varied from 2.04 to 85.57 days. Compared to chlorsulfuron, the degradation rates of L-series target compounds accelerated by 1.84–77.22-fold. The data showed that, for an acyl-substituted amino group at the five positions on the benzene ring, such as in **L101** and **L106**, which had DT_50_ of 2.04 and 68.63 days, respectively, the degradation rates accelerated by 77.22- and 2.29-fold compared to the reference chlorsulfuron. Other than these, the degradation of **L102** (DT_50_ = 15.61 days), which had a cyano group substitution, accelerated 10.09-fold. The DT_50_ of **L103**, which had an iodine substitution, was 27.84 days, and it became 5.66× faster than that of the control. When ethyl, isopropyl, or methyl groups were substituted, such as in **L104**, **L105**, and **L107**, their DT_50_ values became 48.47, 58.25, and 85.57 days, respectively.

Chlorsulfuron is very persistent in alkaline soil. Fredrickson et al., reported that the DT_50_ of chlorsulfuron in silty clay loam was 10 weeks [11]. Thirunarayanan et al., reported that the DT_50_ of chlorsulfuron was 136.6 days in soil with pH 7.7 at 10 °C [15].

In 1999, Singles et al., reported that flupyrsulfuron-methyl (DPX-KR-459) could degrade rapidly at pH 5–9, with a DT_50_ of 0.42–44 days [21]. Villaverde et al., reported that the DT_50_ of flupyrsulfuron-methyl was 5.6 days in soil with pH 7.9, and 9.4 days at pH 8.4 [22]. Rouchaud et al., found that iodosulfuron-methyl degraded rapidly at pH 8.0, with a DT_50_ of 30-40 days [23]. Tang et al., reported that the DT_50_ of iodosulfuron-methyl was 6–23.9 days at pH 5.14–9.42 [24]. Wu et al., reported that the DT_50_ of foramsulfuron was 10.8–31.5 days in soil with pH 5.29–7.86 at 25 °C [25]. Compared with chlorsulfuron, these sulfonylurea herbicides containing fifth substituents on the benzene ring exhibited faster degradation rates.

In 2016, Hua selected the acidic soil from Jiangxi (pH 5.41) with organic matter content 6.85 g·kg^−1^ to investigate the degradation of 5-substituted chlorsulfuron derivatives [12,13]. The DT_50_ values of **L101**–**L107** in a previous study in acidic conditions are listed for comparison in Table 4.

As the data show, the DT_50_ values of **L102**, **L103**, and **L106** were nearly 1.13–2.52-fold slower compared to chlorsulfuron (12.91 days) in acidic soil. In alkaline soil, it was noted that the DT_50_ of **L102**, **L103**, and **L106** were 15.61, 27.84, and 68.63 days, respectively. These compounds indicated an acceleration of 10.09-, 5.66-, and 2.29-fold compared to chlorsulfuron (157.53 days). The degradation rates of these target compounds in alkaline soil were not consistent with those in acidic soil. For **L101**, **L104**, **L105**, and **L107**, the DT_50_ values were nearly 1.16–1.64-fold faster than chlorsulfuron (12.91 days) in acidic soil. However, faster degradation was recorded for these electron-donating compounds in alkaline soil. For instance, **L101** (DT_50_ 2.04 days) accelerated by 77.22-fold compared to chlorsulfuron (DT_50_ = 157.53 days); **L104** (DT_50_ = 48.47 days) degraded 3.25-fold faster; **L105** (DT_50_ = 58.25 days) degraded 2.70-fold faster; and **L107** (DT_50_ = 85.57 days) degraded 1.84-fold faster. The degradation principle of these compounds in alkaline soil was similar to those in acidic soil.

It was found that the introduction of electron-donating groups to benzene rings on their fifth position could hasten their degradation in acidic soil, as well as in alkaline soils. The DT_50_ of this series of compounds was reduced to 2.02–85.57 days, i.e., was accelerated by 1.84–77.22-fold compared to chlorsulfuron. Compared with the DT_50_ of 5-dialkylamino-substituted chlorsulfuron derivatives in alkaline soil [18], this research highlighted that a rational and controllable degradation curve could be achieved by these target compounds.

Based on the small number of tested structures, more structures need to be synthesized to verify whether the degradation patterns of different substituents in alkaline soil are consistent with those in acidic soil. More electron-donating and -withdrawing substituents must be further tested to promote controllable degradation, such as nitro and carboxyl substituents, aliphatic substituents containing halogens, and alkoxy substituents. Additionally, the degradation products of various chlorsulfuron derivatives should be studied in the future. The degradation mechanism might be explored on this basis, as well.

In combination with the bioassay activity, the acidic [12,13] and alkaline soil degradation results were used to conclude that compounds with structures, such as **L104**, **L105**, and **L107**, could potentially be employed as sulfonylurea herbicides. These results can be expected to guide the design of new herbicides with controllable degradation rates in soil through the degradation mechanism.

### 3.2. Crop Safety Results

In this research, the crop safety of target compounds on wheat is listed in Table 5 and Table 6. In addition, the crop safety on corn is listed in Table 7 and Table 8.

As the data presented in Table 5 show, the inhibition rate of **L101** was 1.1% at 30 g·ha^−1^ through the pre-emergence treatment, and the inhibition rate of chlorsulfuron was 0. However, during the post-emergence treatment, the inhibition rate of **L101** was 28.8%, which inhibited the normal growth of wheat.

As the data show in Table 6, the inhibition rates of **L102**–**L107** were 0% through the pre-emergence treatment at 30 g·ha^−1^, which were the same as that of chlorsulfuron (0%). In addition, the inhibition rates of **L103**, **L105**, **L106**, and **L107** on wheat were 0%, 3.9%, 3.7%, and 5.7%, which were better than chlorsulfuron (13.9%). It was speculated that the target compounds could maintain similar crop safety to chlorsulfuron.

As the data show in Table 7, the inhibition rate of **L101** was 0.4% at 30 g·ha^−1^ through the pre-emergence treatment, which was better than chlorsulfuron (32.6%).

As the data show in Table 8, it was noted that the inhibition rates of **L102**, **L104**, **L106**, and **L107** were 0 at 30% and 60 g·ha^−1^ through the pre-emergence treatment, whereas the inhibition rates of chlorsulfuron were 76.1 and 79.4%, respectively. The inhibition rates of **L103** were reduced to 2.3 and 8.4%, respectively, while those of **L105** were 4.1 and 4.6%. For post-emergence treatment, at 30 g·ha^−1^, the inhibition rates of **L102**–**L107** were 15.3, 24.5, 6.7, 4.9, 4.6, and 1.1% compared to chlorsulfuron (45.3%). It was speculated that the five-position substitution could improve the crop safety of sulfonylurea herbicides on corn. In addition, they had harmful effects on corn through post-emergence treatment.

Chlorsulfuron is a classical sulfonylurea herbicide used in wheat fields, but it can seriously endanger the normal growth of subsequent corn [26]. Bayer reported that iodosulfuron-methyl and foramsulfuron are safe for wheat and corn [27,28]. These herbicides contain fifth substituents on the benzene ring, which suggests that fifth substituents on the benzene ring are potential sulfonylurea herbicides, which may improve crop safety. More 5-substituted chlorsulfuron derivatives need to be synthesized and investigated to determine their impact on crop safety. Additionally, different wheat and corn crop species need to be tested.

Based on the above results, we found that five-position substitutions could not only maintain similar crop safety to chlorsulfuron, but they could also improve the crop safety of sulfonylurea herbicides on corn. It was concluded that compounds such as **L104** and **L107** are potential green sulfonylurea herbicides for pre-emergence treatment on wheat and corn.

## 4. Conclusions

Followed by our previously reported studies on herbicidal activities and acidic soil degradation, we systematically studied the degradation and crop safety of 5-substituted chlorsulfuron derivatives in alkaline soil (pH 8.39). It was found that 5-substituted chlorsulfuron derivatives could accelerate degradation in alkaline soil and the DT_50_ of the target compounds by 1.84–77.22-fold compared with chlorsulfuron. This research highlighted that a rational and controllable degradation curve could be achieved by 5-substituted chlorsulfuron derivatives. Additionally, the 5-substituted chlorsulfuron analogues exhibited good crop safety for both wheat and corn through pre-emergence treatment. Moreover, it was noted that the 5-substituted chlorsulfuron compounds could improve the crop safety on corn through post-emergence treatment. In combination with bioassay activities, acidic and alkaline soil degradation, and crop safety, it was concluded that compounds such as **L104** and **L107** are potential green sulfonylurea herbicides for the pre-emergence treatment on wheat and corn. Our findings provide important information for the further design of new sulfonylurea herbicides with high herbicidal activity, rational controllable degradation rates, and high crop safety.

## Figures and Tables

**Figure 1 molecules-27-03318-f001:**
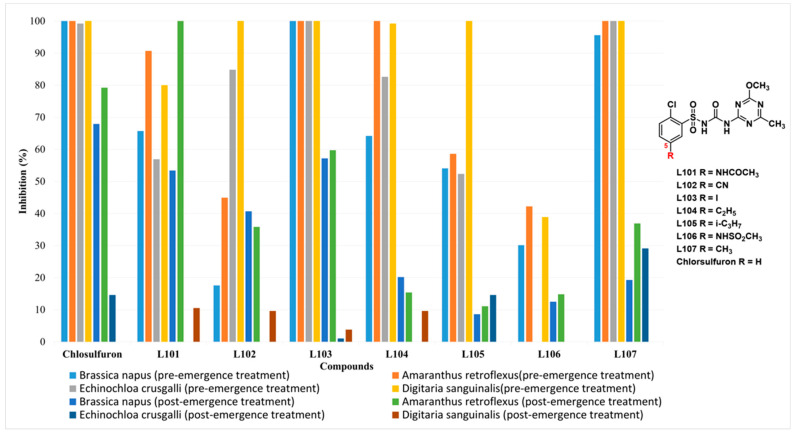
The herbicidal activity of 5-substituted chlorsulfuron compounds against both dicotyledons and monocotyledons at 30 g·ha^−1^. (The full data can be found in Appendix A).

**Figure 2 molecules-27-03318-f002:**
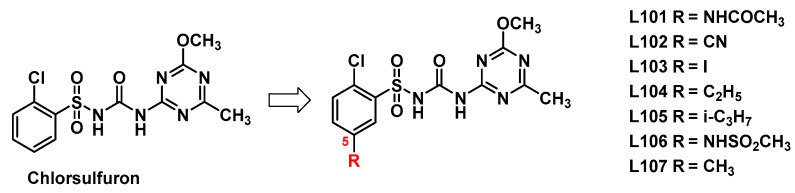
The structure of 5-substituted chlorsulfuron compounds.

**Figure 3 molecules-27-03318-f003:**
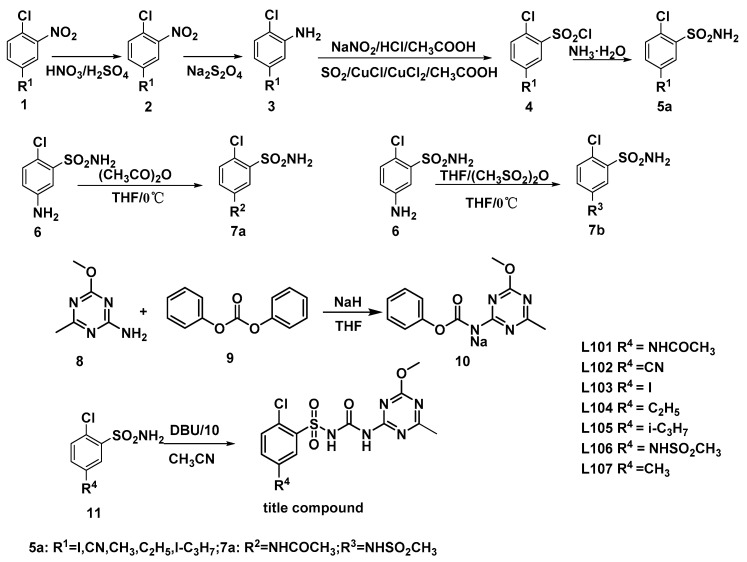
The procedures for synthesising 5-substituted chlorsulfuron compounds.

**Table 1 molecules-27-03318-t001:** Analytical data of soils.

Soils	Soil Texture	pH	Cation Exchange Capacity (cmol^+^·kg^−1^)	Organic Matter (g·kg^−1^)	Soil Separation (mm)/Mechanical Composition (%)								
Alkaline soils	Loam	8.39	7.30	19.4	1–2	0.5–1	0.025–0.5	0.05–0.02	0.02–0.002	<0.002	0.25–0.05	2.0–0.05	0.05–0.002
					0.795	2.46	2.33	7.90	28.6	28.2	29.7	35.3	36.5

**Table 2 molecules-27-03318-t002:** Analytical data on the recovery rates of three concentrations (in soil with pH 8.39).

Compound	HPLC Analysis Condition (Retention Time, Flow Rate, Mobile Phase (*V*:*V*))	Extraction Solvent (*V*:*V*)	Additive Concentration (mg·kg^−1^)	Average Recovery Rate (%)	Coefficient of Variation RSD (%)
**L101**	12.32 min, 0.70 mL·min^−1^, CH_3_OH: H_3_PO_4_ (aq) (pH 3.0) = 64:36	CH_3_COCH_3_:CH_2_Cl_2_:THF: MeOH:H_3_PO_4_ (aq) (pH 1.5) = 30:10:20:20:5	5	82.19	1.04
			2	83.05	2.49
			0.5	81.06	1.81
**L102**	13.01 min, 0.70 mL·min^−1^, CH_3_OH: H_3_PO_4_ (aq) (pH 3.0) = 64:36	CH_3_COCH_3_:CH_2_Cl_2_:THF: MeOH:H_3_PO_4_ (aq) (pH 1.5) = 30:10:10:20:10	5	86.56	1.10
			2	84.17	0.72
			0.5	91.33	1.80
**L103**	12.06 min, 0.80 mL·min^−1^, CH_3_OH: H_3_PO_4_ (aq) (pH 3.0) = 75:25	CH_3_COCH_3_:CH_2_Cl_2_:THF: MeOH:H_3_PO_4_ (aq) (pH 1.5) = 30:10:10:20:10	5	82.21	2.21
			2	82.67	2.14
			0.5	90.01	2.01
**L104**	11.93 min, 0.80 mL·min^−1^, CH_3_OH:H_3_PO_4_ (aq) (pH 3.0) = 75:25	CH_3_COCH_3_:CH_2_Cl_2_:THF: H_3_PO_4_ (aq) (pH 1.5) = 30:10:10:10	5	92.30	2.11
			2	85.51	0.92
			0.5	79.95	2.65
**L105**	15.27 min, 0.80 mL·min^−1^, CH_3_OH: H_3_PO_4_ (aq) (pH 3.0) = 62:38	CH_3_COCH_3_:CH_2_Cl_2_: H_3_PO_4_ (aq) (pH 1.5) = 40:5:5	5	85.10	1.74
			2	86.91	0.68
			0.5	90.14	1.59
**L106**	14.48 min, 0.75 mL·min^−1^, CH_3_OH: H_3_PO_4_ (aq) (pH 3.0) = 75:25	CH_3_COCH_3_:CH_2_Cl_2_:MeOH: H_3_PO_4_ (aq) (pH 1.5) = 30:10:30:10	5	82.40	1.37
			2	82.62	0.99
			0.5	74.46	2.99
**L107**	12.32 min, 0.80 mL·min^−1^, CH_3_OH: H_3_PO_4_ (aq) (pH 3.0) = 62:38	CH_3_COCH_3_:CH_2_Cl_2_:THF: MeOH:H_3_PO_4_ (aq) (pH 1.5) = 30:10:10:10:10	5	87.03	1.21
			2	88.33	0.68
			0.5	78.70	1.13
**Chlorsulfuron**	12.66 min, 0.70 mL·min^−1^, CH_3_OH: H_3_PO_4_ (aq) (pH 3.0) = 62:38	CH_3_COCH_3_:CH_2_Cl_2_:H_3_PO_4_ (aq) (pH 1.5): = 40:5:10:10	5	73.54	1.09
			2	73.53	2.40
			0.5	81.09	1.16

**Table 3 molecules-27-03318-t003:** Kinetic parameters for alkaline soil (pH 8.39) degradation.

**Compound**	Kinetic Equations of Soil Degradation	Correlation Coefficient (R^2^)	DT_50_ (Days)
**L101**	*C_t_* = 4.5066e^−0.3406t^	0.9888	2.04
**L102**	*C_t_* = 4.5113e^−0.0444t^	0.9922	15.61
**L103**	*C_t_* = 4.9924e^−0.0249t^	0.9960	27.84
**L104**	*C_t_* = 4.7036e^−0.0143t^	0.9983	48.47
**L105**	*C_t_* =4.2712e^−0.0119t^	0.9977	58.25
**L106**	*C_t_* = 3.7330e^−0.0101t^	0.9926	68.63
**L107**	*C_t_* = 4.3097e^−0.0081t^	0.9964	85.57
**Chlorsulfuron**	*C_t_**=* 4.3067e^−0.0044t^	0.9884	157.53

**Table 4 molecules-27-03318-t004:** Comparison of acid and alkaline soil degradation results of target compounds.

Compd.	DT_50_ (Days)	
	Acidic Soil (pH = 5.41)	Alkaline Soil (pH = 8.39)
**L101**	10.76	2.04
**L102**	32.54	15.61
**L103**	14.78	27.84
**L104**	7.89	48.47
**L105**	9.40	58.25
**L106**	14.62	68.63
**L107**	11.16	85.57
**Chlorsulfuron**	12.91	157.53

**Table 5 molecules-27-03318-t005:** Crop safety of compound **L101** on wheat.

Compound	Concentration (g·ha^−1^)	Wheat (Xinong 529)							
		Pre-Emergence (22 Days after Treatment)				Post-Emergence (28 Days after Treatment)			
		Fresh Weight g/10 Strains	Analysis of Variance ^a^		Inhibition (%)	Fresh Weight g/10 Strains	Analysis of Variance ^a^		Inhibition (%)
			5%	1%			5%	1%	
	0	2.423	ab	A		2.998	a	A	
**Chlorsulfuron**	30	2.411	ab	A	0	2.576	ab	AB	14.1
	60	2.519	a	A	0.5	2.538	abc	AB	15.3
**L101**	30	2.396	ab	A	1.1	2.133	bcd	ABC	28.8
	60	2.300	ab	A	5.1	2.117	bcde	ABC	29.4

^a^ Among the averages, the same letter indicates that there was no significant difference, and different letters indicate that there was a significant difference.

**Table 6 molecules-27-03318-t006:** Crop safety of compounds **L102**–**L107** on wheat.

Compound	Concentration (g·ha^−1^)	Wheat (Xinong 529)							
		Pre-Emergence (22 Days after Treatment)				Post-Emergence (28 Days after Treatment)			
		Fresh Weight g/10 Strains	Analysis of Variance ^a^		Inhibition (%)	Fresh Weight g/10 Strains	Analysis of Variance ^a^		Inhibition (%)
			5%	1%			5%	1%	
	0	2.405	a	A		3.286	abc	ABC	
**Chlorsulfuron**	30	2.421	a	A	0	3.691	bcde	ABC	13.9
	60	2.438	a	A	0	3.661	bcde	BC	14.6
**L102**	30	2.560	a	A	0	3.201	de	BCD	25.3
	60	2.224	a	A	7.5	3.140	de	BCD	26.7
**L103**	30	2.404	a	A	0	4.471	ab	AB	0
	60	2.450	a	A	0	5.058	a	A	0
**L104**	30	2.422	a	A	0	3.585	bcde	BC	16.4
	60	2.585	a	A	0	3.523	bcde	BC	17.8
**L105**	30	2.589	a	A	0	4.120	abcd	ABC	3.9
	60	2.483	a	A	0	3.794	bcde	ABC	11.5
**L106**	30	2.464	a	A	0	4.127	abcd	ABC	3.7
	60	2.472	a	A	0	3.893	bcde	ABC	9.2
**L107**	30	2.480	a	A	0	4.043	bcd	ABC	5.7
	60	2.449	a	A	0	3.945	bcde	ABC	7.9

^a^ Among the averages, the same letter indicates that there was no significant difference, and different letters indicate that there was a significant difference.

**Table 7 molecules-27-03318-t007:** Crop safety of target compound **L101** on corn.

Compound	Concentration (g·ha^−1^)	Corn (Xindan 66)							
		Pre-Emergence (16 Days after Treatment)				Post-Emergence (23 Days after Treatment)			
		Fresh Weight g/5 Strains	Analysis of Variance ^a^		Inhibition (%)	Fresh Weight g/5 Strains	Analysis of Variance ^a^		Inhibition (%)
			5%	1%			5%	1%	
	0	11.599	a	AB		9.214	a	A	
**Chlorsulfuron**	30	7.813	b	BC	32.6	5.928	bc	BCD	35.7
	60	4.463	c	C	61.5	4.771	bcde	BCD	48.2
**L101**	30	11.548	a	AB	0.4	5.146	bcde	BCD	44.1
	60	10.949	ab	AB	5.6	4.291	cde	BCD	53.4

^a^ Among the averages, the same letter indicates that there was no significant difference, and different letters indicate that there was a significant difference.

**Table 8 molecules-27-03318-t008:** Crop safety of target compounds **L102**–**L107** on corn.

Compound	Concentration (g·ha^−1^)	Corn (Xindan 66)							
		Pre-Emergence (16 Days after Treatment)				Post-Emergence (23 Days after Treatment)			
		Fresh Weight g/5 Strains	Analysis of Variance ^a^		Inhibition (%)	Fresh Weight g/5 Strains	Analysis of Variance ^a^		Inhibition (%)
			5%	1%			5%	1%	
	0	8.302	ab	AB		14.413	a	A	
**Chlorsulfuron**	30	1.982	e	D	76.1	7.883	cd	B	45.3
	60	1.713	e	D	79.4	7.796	cd	B	45.9
**L102**	30	9.117	a	A	0	12.213	abc	AB	15.3
	60	8.311	ab	AB	0	10.515	abcd	AB	27.0
**L103**	30	8.109	abc	AB	2.3	10.883	abcd	AB	24.5
	60	7.605	abc	AB	8.4	10.190	abcd	AB	29.3
**L104**	30	8.370	ab	AB	0	13.440	ab	AB	6.7
	60	8.336	ab	AB	0	12.423	abc	AB	13.8
**L105**	30	7.963	abc	AB	4.1	13.708	ab	AB	4.9
	60	7.923	abc	AB	4.6	13.500	ab	AB	6.3
**L106**	30	8.443	ab	AB	0	13.750	ab	AB	4.6
	60	8.508	ab	AB	0	13.700	ab	AB	4.9
**L107**	30	8.335	ab	AB	0	14.250	a	A	1.1
	60	8.327	ab	AB	0	12.668	ab	AB	12.1

^a^ Among the averages, the same letter indicates that there was no significant difference, and different letters indicate that there was a significant difference.

## Data Availability

Data are contained within the article and Appendix A.

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
