# Peer review of "Alkaline Soil Degradation and Crop Safety of 5-Substituted Chlorsulfuron Derivatives"

_molecules, 2022, doi:10.3390/molecules27103318_

Round 1

Reviewer 1 Report

A novel idea with issues on the experimental design itself. A single soil type (loam, pH 8.39), 1 variety of local wheat, 1 variety of local corn. 

Please refer to the text (in yellow). 

The study is acceptable with major revision. The fundamental idea behind the paper is good. 

The issues are as raised and stated in the yellow markers in the pdf text.

  1. Acid soil is cross-referenced without data shown for acid soil in the text.
  2. The study focused on a single alkaline soil at constant pH 8.39 , with a single local variety of wheat and corn. This warrants only local interest, highly.
  3. Figure 4. both caption states alkaline soil. Figure 5. Acid soil? No mention in the text.  
  4. The use of Duncan was clearly stated. Duncan/Tukey/LSD etc. Duncan is often suitable for high variation in data. The variation was not well observed in the text. Besides that, the inclusion of a correlation study can assist in the justification of the results.   
  5. The study assumes a number of ideas with limited justification (pg. 5-9).  

Reviewer 2 Report

Title:

1/ "Rational Controllable" sounds out of context, condensate the main discovery into a short and groundbreaking claim

Abstract:

2/ better address our international audience, do not indicate local impact by referring to any local institution

3/ better explain why/how are your findings are transferable, highlight the urgency and significance from global point of view

Introduction:

4/ deeper review the economical standpoints behind the background of plant production

5/ please understand that the chemical nature of things is global, do not limit your Introduction chapter to China

6/ Fig. 1: do not indicate that activity over 100 % is possible

7/ clearly build the research hypothesis

Materials and Methods:

8/ the method must be presented in such a way that it can be reproduced anytime, by anyone, anywhere (do not create obstacles like referring to specific location etc.)

9/ consider providing cost breakdown or any other financial analysis if you want to argue the research is industrially significant

Results and Discussion:

10/ show more self-criticism to your work (can all the methods and results be fully trusted? what are the weaknesses of the methods used? where do the main measurement inaccuracies arise? what are the limitations from a commercial point of view? are the lessons learned transferable to other fields?)

11/ propose some improvements and direction for future research

12/ reveal the main driving mechanisms of your results, provide deeper synthesis and revel some more original/significant findings

Reviewer 3 Report

In paragraph 4 of the introduction, the first two sentences are not supported by literature citations; please add references. In line 4 of paragraph 4, the author says "In our previous study" and quote Li et al. There is actually no cited reference with Li as first author in the reference section. In line 6 of that paragraph, the author referred to "they" which is confusing.

The rational and motivation does not give strong evidence as to why this study is important for the farming community.

In the method section, 2,3 soil degradation assay. The HPLC conditions are not well described. What standards were used and what was the concentration ranges for the standard curves. Not only methanol and water are used as mobile phase according to table 2. What are the retention time of the different test compound?

It is not necessary to include the wavelength in every block for different compound since it is the same. Please include in the description above.

In the results and discussion section, it is expected to first present the results as they were before discussing them. or have a separate results section. Avoid discussing results before presenting them first.

Reviewer 4 Report

The manuscript is interesting, and this study is necessary, considering the importance of employed new and efficient sulfonylurea herbicides with high activity, adequate degradation rate on certain types of soil and crop safety (wheat and corn). It is understandable, and it is well organized. I find no problems with the scientific approach or technical content presented by the authors in this manuscript.

Round 2

Reviewer 1 Report

An overall good scientific study. 

This manuscript is a resubmission of an earlier submission. The following is a list of the peer review reports and author responses from that submission.

Round 1

Reviewer 1 Report

  • The Introduction only reports on the studied herbicide, but without demonstrating gaps and scientific novelties. The objectives are not hypotheses evaluation proposals, but only experimental protocols.
  •  Methodology presents few details about the protocols and machines used. 
  • The results are poorly presented, with no introduction or context. There is no discussion. 

Author Response

Thank you for your advice. According to your suggestions, we have revised and added some content. Please see the attachment.

Reviewer 2 Report

Dear author(s),
this manuscript brings some inspiring insights to chlorosulfuron degradation in alkaline soil and I tend to quickly agree on its publication. However, there are some issues that can be immediately addressed to improve the overall communication of your work:

Title:

  • condensate the main discovery into a short and groundbreaking claim
  • please understand that the purpose of Title and Abstract is to provide anybody (including experts from other fields) a clear idea what is the paper about, do not use any technical/chemical terms, abbreviations, symbols etc.

Abstract:

  • better address our international audience of readers, make sure the research is globally applicable (not limited to China)
  • follow the established schema of writing academic Abstract: A/ motivation + research hypothesis; B/ methods + results; C/ conclusions and interdisciplinary implications
  • do not refer to any local authorities, bring your work on higher/global level
  • significantly limit the use of technical terms and abbreviations, kindly note that the essence of the Abstract is to communicate to all readers (including those from other disciplines) what the manuscript is about
  • do not present revelations that are limited to your case study, provide deeper synthesis of your findings and present new theoretical revelations (globally applicable/transferable knowledge is missing)
  • it is hard to find something new or unexpected, highlight the scientific novelty and quantify the economic importance of your discovery (clarify how will humanity benefit from your work)

Introduction:

  • be more explanatory and make the manuscript understandable to a wider range of readers, so as to gain more citations, better explain "sulfonylurea", "acetolactate synthase" and all other technical and chemical terms, make sure the Introduction chapter fulfills its purpose (all the abbreviations, symbols and terminology used in the following chapters needs to be fully explained)
  • avoid reference overkill by breaking down all clusters of references (use only 1 reference per claim/sentence)
  • make sure the topic is not limited to China, deeper review the state of art and refer to last papers in the field
  • build your research hypothesis more clearly (straightforward and groundbreaking claim that is confirmable or refutable) at the end of the Introduction chapter, justify the urgency of its investigation from global point of view (explain how will our readers benefit from your work, the urgency and significance should be justified from global point of view)

Materials and Methods:

  • our readers should find here (only and exclusively) detailed description of all your procedures (step by step), describe each apparatus (serial number, manufacturer, country of origin, setup, process parameters etc.), reactant (manufacturer, country of origin, purity) and method used, anybody who reads this chapter should be able to repeat your methods and obtain exactly the same results
  • analysis on quantity of organic matter is not enough, analyze its quality if possible
  • do not ignore economic reality, analyze cost breakdown or at least some simplified financial analysis

Results and Discussion:

  • show more self-criticism to your methods, discuss all limitations of your results (are these results globally reproducible and significant?)
  • each Fig. and Tab. should be provided with detailed caption that will explain A/ what can be seen; B/ why is it important and C/ how is it related to the research hypothesis
  • more intensely compare your findings with existing literature and discuss industrial and multidisciplinary implications
  • anytime you use "/" in any scientific graph or table, you should understand that this symbols means "dividing"
  • take your research to the next level, provide a deeper synthesis of your results and reveal the mechanisms that shape them, this will allow you to uncover original theoretical insights
  • indicate direction for future research and propose some improvements
  • discuss at least some basic economic considerations, comment on competitiveness with existing practice

Conclusion:

  • depersonalize the entire manuscript, remove all "we" etc.
  • please understand that the Conclusion chapter is not the same as the Abstract = not a summary of your work (do not repeat description of your methods and summary of your results again and again), make sure you are presenting only new theoretical findings that originate firstly from your work and are not deducible from other literature
  • synthesis should be followed by generalization (from global point of view)
  • clearly indicate whether your research hypothesis tends to be confirmed or not, highlight/quantify the industrial/environmental significance

Author Response

(The authors gave the same response as above.)

Reviewer 3 Report

  1. The introduction could be more detailed;
  2. No research hypothesis is presented in the introduction;

  3. I recommend presenting the discussion in a separate chapter. The discussion with other authors could have been more detailed;

  4. Few literature sources are analyzed in the article. I recommend not to use old literature sources;

  5. Methods of statistical analysis need to be described more clearly;
  6. In the figures No. 3 and 4 not presented statistical evaluation indicators;

  7. I recommend figure No. 1 moving to the section "Materials and Methods".

Author Response

(The authors gave the same response as above.)

Round 2

Reviewer 1 Report

The authors effectively sought to improve the text according to the suggestions given by this reviewer. However, I consider that the corrections did not meet the expected objectives. The Introduction still does not present a scientific gap-scientific hypothesis axis, the methods need to be more detailed, and the discussion remains poor. I suggest a global re-evaluation of the article. 

Reviewer 2 Report

My comments were poorly understood and badly addressed. 

Unless my comments are addressed or refuted I cannot recommend the manuscript for publication.

Reviewer 3 Report

No comments.